# Race- and Gender-Specific Associations between Neighborhood-Level Socioeconomic Status and Body Mass Index: Evidence from the Southern Community Cohort Study

**DOI:** 10.3390/ijerph20237122

**Published:** 2023-11-30

**Authors:** Lauren Giurini, Loren Lipworth, Harvey J. Murff, Wei Zheng, Shaneda Warren Andersen

**Affiliations:** 1Department of Population Health Sciences, School of Medicine and Public Health, University of Wisconsin-Madison, Madison, WI 53705, USA; lgiurini@wisc.edu; 2Carbone Cancer Center, University of Wisconsin-Madison, Madison, WI 53705, USA; 3Division of Epidemiology, Department of Medicine, Vanderbilt Epidemiology Center, Vanderbilt-Ingram Cancer Center, Vanderbilt University School of Medicine, Nashville, TN 37232, USA; loren.lipworth@vumc.org (L.L.); wei.zheng@vumc.org (W.Z.); 4Department of Medicine, Vanderbilt University Medical Center, Nashville, TN 37232, USA; harvey.j.murff@vanderbilt.edu

**Keywords:** neighborhoods, obesity, social determinants, socioeconomic status, race, disparities, body mass index

## Abstract

Obesity and a low socioeconomic status (SES), measured at the neighborhood level, are more common among Americans of Black race and with a low individual-level SES. We examined the association between the neighborhood SES and body mass index (BMI) using data from 80,970 participants in the Southern Community Cohort Study, a cohort that oversamples Black and low-SES participants. BMI (kg/m^2^) was examined both continuously and categorically using cut points defined by the CDC. Neighborhood SES was measured using a neighborhood deprivation index composed of census-tract variables in the domains of education, employment, occupation, housing, and poverty. Generally, the participants in lower-SES neighborhoods were more likely to have a higher BMI and to be considered obese. We found effect modification by race and sex, where the neighborhood-BMI association was most apparent in White female participants in all the quintiles of the neighborhood SES (OR_Q2_ = 1.55, 95%CI = 1.34, 1.78; OR_Q3_ = 1.71, 95%CI = 1.48, 1.98; OR_Q4_ = 1.76, 95%CI = 1.52, 2.03; OR_Q5_ = 1.64, 95%SE = 1.39, 1.93). Conversely, the neighborhood-BMI association was mostly null in Black male participants (OR_Q2_ = 0.91, 95%CI = 0.72, 1.15; OR_Q3_ = 1.05, 95%CI = 0.84, 1.31; β_Q4_ = 1.00, 95%CI = 0.81, 1.23; OR_Q5_ = 0.76, 95%CI = 0.63, 0.93). Within all the subgroups, the associations were attenuated or null in participants residing in the lowest-SES neighborhoods. These findings suggest that the associations between the neighborhood SES and BMI vary, and that other factors aside from the neighborhood SES may better predict the BMI in Black and low-SES groups.

## 1. Introduction

Epidemiological studies have documented an association between the neighborhood-level socioeconomic status (SES) and body mass index (BMI), where residing in a neighborhood with a lower SES is associated with a higher BMI and obesity [1,2,3,4]. The neighborhood-level SES represents the social and economic conditions of a place constructed from area-level measures of poverty, education, employment, occupation, and housing [5]. Additionally, the neighborhood-level SES is often correlated with attributes of the built environment, where lower-SES neighborhoods have fewer parks, trails, and grocery stores [6,7,8,9]. Less access to green space and fresh-food grocery stores, measured at the individual level, is associated with an increased risk of obesity [10,11,12,13].

The present study explores the relations between the neighborhood-level SES and BMI with a specific focus on overweight and obesity. Due to disparities in obesity by race and gender [14,15], we assessed the potential that the impact of the neighborhood-level SES may be more apparent in certain demographic subpopulations. In support of this assertion, previous studies have documented that the neighborhood may more strongly influence the risk of obesity in women as compared to men [16,17,18], possibly due to differences in the importance of social networks, support, and neighborhood safety [16,19]. Moreover, residential segregation and discrimination may also play a role in the social cohesiveness of neighborhoods, and these issues are more often experienced by Black individuals than other racial groups [20,21]. The Southern Community Cohort Study (SCCS) is a prospective cohort study that provides a rare opportunity to study the association between neighborhood-level and individual-level exposures among individuals that are of a low individual-level and neighborhood-level SES. The findings from the present study will extend previous research by including populations that have typically been underrepresented in research.

## 2. Materials and Methods

### 2.1. Data Collection

The data arose from the Southern Community Cohort Study (SCCS), on which details have been previously published [22,23]. The study enrolled approximately 83,485 English-speaking participants, aged 40–79 years old, from the southeastern United States between 2002 and 2009. The majority of the enrollment took place at community health centers, which allowed for an oversampling of individuals with a low socioeconomic status. The remaining cohort was enrolled via a mailed questionnaire sent to a stratified random sample of residents in the same 12 geographic states. Upon entry, the participants completed a baseline survey that collected information about their self-identified gender, race, ethnicity, lifestyle, residential address, and familial and individual health history. The SCCS was approved by IRB at the Vanderbilt University Medical Center and Meharry Medical College; all participants provided their written consent for participation.

### 2.2. Exposure Measurement

The previously described neighborhood deprivation index [5,23] was used as a proxy for the neighborhood-level SES, measured at the census-tract level, and linked with participants’ addresses at the time of the baseline interview. The index was constructed through a principal component analysis of 11 variables from the 2000 census in the domains of education, employment, income, car ownership, home ownership, home crowding, households receiving public assistance, and prevalence of poverty. The values reflected socioeconomic disadvantages relative to other census tracts in the 12 southeastern United States. Most participants in the SCCS had an address in the quintile with the highest deprivation, likely due to the majority of the sampling taking place at community health centers.

### 2.3. Outcome Measurement

The BMI (kg/m^2^) was collected via the self-reported height and weight at the time of study entry. The cutoffs for BMI categories were determined by the designations for underweight (<18.5), “healthy” weight (18.5–24.9), overweight (25–29.9), and obese (≥30.0), as defined by the CDC [24].

### 2.4. Participants

Participants with missing data on the neighborhood deprivation index (n = 1212) or body mass index (n = 1354) were excluded from these analyses. The data from 80,970 SCCS participants were available for analysis after these exclusions. Missing values for the covariates sex, race, enrollment age, education, insurance coverage, household income, and smoking status were imputed using the sex- and race-specific mode or median.

### 2.5. Statistical Analysis

Frequency distributions for the neighborhood deprivation index quintile and BMI category were tabulated. The continuous mean BMI was estimated with linear regression. Additionally, odds ratios (ORs) and 95-percent confidence intervals (CIs) of being underweight, overweight, or obese compared to “healthy”-weight participants were estimated using logistic regression. The statistical models included an adjustment for the covariates sex (female or male), race (non-Hispanic Black, non-Hispanic White, Hispanic, or other), enrollment age (40–44, 45–49, 50–54, 55–59, 60–64, 65–69, or 70+), education (less than 9 years; greater than 9, but less than 11 years; high school diploma or GED; vocational or technical degree or some college; or college degree or more), insurance coverage (yes or no), household income (less than or equal to $14,999; $15,000–24,999; $25,000–49,999; $50,000–99,999; or $100,000 or more), smoking status (current, former, or never), and enrollment source (community health center or other). Likelihood ratio tests were used to determine the potential interactions between the neighborhood deprivation index and covariates of interest in association with overweight/obesity.

## 3. Results

At the time of study enrollment, the majority of the participants in this cohort were of non-Hispanic Black race, had a household income <$15,000, did not have educational attainment beyond a high school degree, were current or former smokers, and lived in the most deprived quintile of the neighborhood SES (Table 1). The neighborhood SES was associated separately with BMI, obese status (BMI > 30) and overweight status (BMI 25–29.9) (see Table 2; Figure 1; and Appendix A). The associations varied by sex and race (*p*-interactions < 0.01). There was a null association between the neighborhood SES and an underweight status (BMI < 18.5) (OR = 1.03, 95%CI = 0.97, 1.09); therefore, the underweight estimates were omitted from further analyses.

The neighborhood-level SES was associated with BMI most notably in non-Hispanic White males and females (see Table 2). Within non-Hispanic White males, living in a lower-SES neighborhood was associated with a heightened BMI in quintiles for the second- through the fourth-highest SES neighborhoods. In White females, living in a lower-SES neighborhood was also associated with a heightened BMI, where there was a gradual increase in the average BMI from the residents of the second- to the fourth-highest SES neighborhoods (β_Q2_ = 1.56, 95%SE = 0.22; β_Q3_ = 1.72, 95%SE = 0.22; β_Q4_ = 2.02, 95%SE = 0.22). Slight increases in the BMI were also apparent in non-Hispanic Black females in the third- through the fifth-highest quintiles of neighborhood SES. No association was seen between the neighborhood SES and BMI in non-Hispanic Black male participants.

A lower neighborhood-level SES was associated with obesity, with the associations being most apparent among non-Hispanic White females (see Appendix A, Figure 1). Among non-Hispanic White female participants, residing in a lower-SES neighborhood was consistently associated with an increased obesity prevalence. For the other demographic subpopulations, non-linear associations were observed where, in comparison to “healthy” weight participants that resided in the neighborhoods with the highest SES, participants in the quintiles for the second- through the fourth-highest SES neighborhoods were often at an increased risk of obesity (White males: OR_Q3_ = 1.45, 95%CI = 1.20, 1.76; Black females: OR_Q3_ = 1.43, 95%CI = 1.18, 1.74), and the association was attenuated or null for the participants that resided in the quintile for the lowest neighborhood-level SES (White males: OR_Q5_ = 1.03, 95%CI = 0.84, 1.25; Black females: OR_Q5_ = 1.15, 95%CI = 0.98, 1.36). There was no association between the neighborhood-level SES and obesity among non-Hispanic Black males.

Within the race–gender strata, associations between the neighborhood SES and an obese status were modified by the insurance status and individual-level income. Specifically, a modification by insurance status was noted among White males (*p*-interaction = 0.009) and non-Hispanic Black females (*p*-interaction = 0.05), where those with insurance were more likely to be obese than those without (see Appendix A).

Modification by individual-level income was observed among non-Hispanic Black females (*p*-interaction = 0.001). A lower neighborhood-level SES was associated with increased odds of obesity among non-Hispanic Black female participants with >$50,000 in household income, whereas there was a null association between the neighborhood-level SES and obesity among non-Hispanic Black female participants with <$15,000 in household income (see Figure 2 and Appendix A). A difference in the association by household income was also evident among non-Hispanic Black male participants; however, all confidence intervals crossed unity.

Obesity was more common among females than men, and the majority of non-Hispanic Black females enrolled were considered obese (see Figure 3). Additionally, among non-Hispanic White participants, there was a larger spread of neighborhood-level SES, whereas the majority of non-Hispanic Black participants resided in the fifth quintile, which represented the lowest neighborhood-level SES (see Figure 3).

The neighborhood-level SES was also associated with overweight status, with the largest associations amongst non-Hispanic White females (see Appendix A). In non-Hispanic White female participants, the neighborhood SES was consistently inversely associated with having an overweight BMI. In other demographic groups, the associations were inconsistent across quintiles; however, in comparison to “healthy”-weight participants that resided in the neighborhoods with the highest SES, the participants in the quintiles for the second- and third-highest SES were commonly at the highest risk of being overweight (White males: OR_Q3_ = 1.23, 95%CI = 1.04, 1.48; Black females: OR_Q3_ = 1.22, 95%CI = 0.98, 1.51). This association was attenuated or null for participants that resided in the quintile for the lowest neighborhood-level SES (White males: OR_Q2_ = 1.03, 95%CI = 0.86, 1.24; Black females: OR_Q3_ = 1.01, 95%CI = 0.84, 1.21). No association between the neighborhood SES and an overweight BMI was present in non-Hispanic Black males. Within the race–gender strata, no effect modification was observed for the association between the neighborhood-level SES and overweight status by insurance, income, or smoking status (all *p*-interactions > 0.08).

## 4. Discussion

This prospective cohort study provides evidence that the neighborhood-level SES is associated with overweight and obese status in a cohort of adults who reside in highly deprived SES environments; these associations were the greatest in non-Hispanic White women. These results are in line with previous studies of the association between neighborhood socioeconomic indices and obesity [1,2,3,4]. However, ours is the first to conduct this research in a cohort that oversamples non-Hispanic Black and low individual-level SES populations. Most of our participants resided in the quintile for the lowest neighborhood-level SES, indicating that the majority of the participants lived in a census tract with low graduation rates, where few residents own their own home, or car, or make more than $30,000 in income. In these census tracts, it is more common to be living in poverty, receiving public assistance, or both [25]. This contrasts with the highest-SES quintile, where most residents are high school graduates, own their home, own a car, and make more than $30,000 in income, and few households live beneath the poverty line or receive public aid [25]. Since the majority of the SCCS cohort lived in the quintile of the lowest neighborhood SES, it allowed us to extend the field’s knowledge to those typically underrepresented in epidemiologic cohort studies.

Our findings suggest that non-Hispanic White female participants are more vulnerable to the influences of neighborhood SES than their male counterparts. While part of this finding may be due to the differences in the prevalence of exposure and outcome by race and sex, other studies provide concurrent evidence. Letarte et al. found a modification by sex in the association between cross-sectional and cumulative exposures to neighborhood deprivation and obesity [17]. Fan et al. also reported an association between the census-tract SES and obesity/BMI in females, where the relationship was null in males [18]. Moreover, Cuevas et al. documented an association between financial, relationship, and cumulative stressors and BMI in females, but not males [26]. Possible explanations for the sex difference may be related to preferences for social support and differing levels of social cohesion by neighborhood-level SES [16,18]. Ours and the previously published results provide support for customizing or targeting interventions in the neighborhood socioeconomic environment towards females.

In addition, our results show that the association between the neighborhood-level SES and obesity is less strong in participants who reside in the lowest-neighborhood-level-SES areas (quintile 5). One explanation may be that neighborhood resources are less important to health than other factors for very socioeconomically vulnerable participants. We hypothesize that, for non-Hispanic Black males, part of the explanation may be the lower prevalence of obesity in comparison to “healthy” and overweight people. Additionally, the majority of non-Hispanic Black male participants (57%) were current smokers; previous studies have linked current smoking with a decreased BMI [27,28].

Previous differences by race have been found in the association between individual-level sociodemographic, social, and built-environment characteristics and obesity [29]. While our results are novel methodologically, they do agree with the work by Cuevas et al. documenting an association between the neighborhood-level financial stress and BMI in White subjects [26]. Cuevas et al. found that the association between a neighborhood stress score, which measured safety, vandalism, and community disorder, with BMI varied by race. They found that White participants are more likely to have an increased BMI and Black participants are more likely to have a reduced BMI associated with a neighborhood stress when adjusting for individual-level smoking [26]. We found similar associations when comparing non-Hispanic White participants to non-Hispanic Black participants, specifically in Black men in the lowest-SES quintile. We hypothesize that, in Black participants, other factors may play a more important role in determining BMI than the neighborhood SES. Specifically, the association between the neighborhood-level SES and BMI was attenuated among participants with fewer resources, as measured by the individual-level income and health insurance status. For Black female participants, we found a modification by household income, where the association between the neighborhood-level SES and BMI was more apparent among participants with a household income greater than $50,000 than those with less than $15,000 in household income. Previous studies have documented an interaction between the neighborhood SES and the household SES in the risk of obesity in children, not adults [30,31,32]. One potential explanation may be that individual-level income is a greater determinant of one’s caloric intake and expenditure than neighborhood-level SES. Moreover, with respect to our insurance status results, possible explanations for a stronger relation between the neighborhood-level SES and BMI among the insured is that these participants held positions that not only provided them insurance coverage, but also afforded them more sedentary time, more caloric intake, and modes of less active transportation. In both the modifications by income in Black females and insurance in White males, we found that the less privileged groups had a null association between the neighborhood-level SES and BMI. This finding is interesting, as it points to the possibility that fewer resources on an individual level may blunt the effects of the neighborhood environment.

Part of the observed differences in the neighborhood-level SES were related to redlining, a historical practice of discriminatory mortgage-lending established in the 1930s by the Home Owner’s Loan Association. Mortgage lenders would refuse loans to people of color for homes in more affluent, White neighborhoods, which resulted in the clustering of racial minorities in more deprived neighborhoods, where there are still higher levels of pollution, crime, and policing and less access to healthy food sources today [33,34,35]. This makes studying the effects of the neighborhood-level SES on health in the SCCS important in understanding the long-term effects of discriminatory policies.

## 5. Strengths and Limitations

The Southern Community Cohort Study (SCCS) is a large, prospective, and diverse cohort. The majority of participants are of Black race and have a low income, allowing researchers to study populations that are typically excluded from research even though they have the highest prevalence of disease. This study, and others examining the determinants of BMI, are important not only due to the increasing burden of obesity in the United States, but also due to the nature of obesity as a precursor to a variety of other conditions. Our results show that policies to change the socioeconomic component of neighborhoods could reduce the obesity burden.

Our study also has certain limitations. First, we note that the participants may have self-selected or had access to specific neighborhoods based on a combination of race, resources, and preferences [36]. We also acknowledge that our measurement of the neighborhood socioeconomic environment may not accurately predict every participant’s neighborhood experience. Research has shown that residents may view their neighborhood as smaller than a census tract [37], and therefore, we may not have captured all the variation that occurs within a tract. Further, the neighborhood deprivation index was calculated from values from the 2000 census, which was linked to participants’ addresses reported at their baseline interview in 2002–2009, so there is possible error in this measure, as neighborhoods change over time. With regard to our outcome measurement, we note that the BMI is imperfect in measuring excess body weight; however, this is particularly an issue for highly muscular individuals, which are especially rare in this cohort. Finally, we acknowledge that the traditional BMI cutoffs may not accurately measure obesity among racial and ethnic minorities, and due to this, we have also included analyses of the BMI as a continuous outcome. Despite these shortcomings, the fact that our neighborhood-level SES measure was composed using 11 census-tract variables increases its validity, allowing for a good approximation of a participants’ neighborhood-level exposure to wealth, and the inclusion of the BMI as a continuous predictor may account for errors in the categorical assignment.

## 6. Conclusions

Overall, this study documents that neighborhood socioeconomic deprivation increases the likelihood of obesity, and that the association between the neighborhood SES and BMI varies by race and sex, with the strongest associations occurring in those identifying as non-Hispanic White females. These findings are critical because obesity increases an individual’s risk for a multitude of diseases as well as a lower life expectancy. Further, herein, we provide support for targeted interventions, such as increasing nutrition education in schools and structural investments, such as the construction of a park or bike path, in low-SES communities in order to minimize health disparities.

## Figures and Tables

**Figure 1 ijerph-20-07122-f001:**
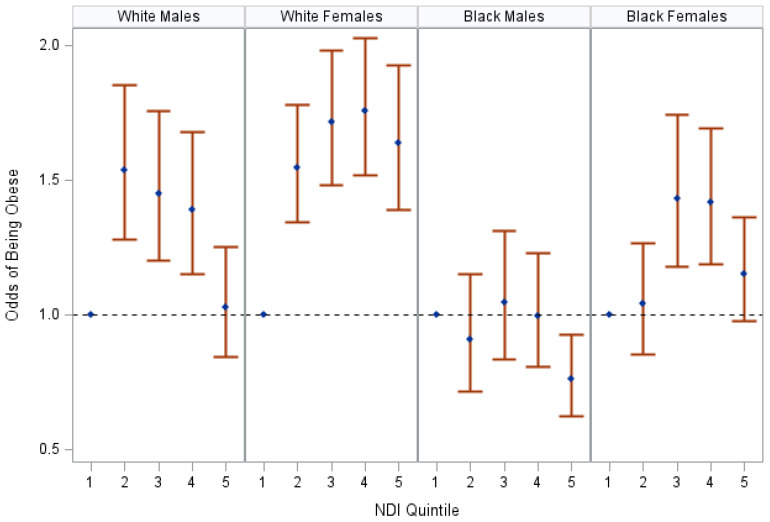
The association between NDI and obese status, stratified by race and sex, SCCS at baseline, 2002–2009. The figure displays odds ratios and 95% confidence intervals of being obese at the time of the participants’ baseline interviews in reference to “healthy”-weight participants living in a neighborhood in the highest-SES quintile (Q1). Odds ratios were adjusted for age, insurance coverage, education, household income, and smoking status. Neighborhood SES is represented by the neighborhood deprivation index (NDI); “healthy weight” is defined by having a body mass index (BMI) of (18.5–24.9); obese status is defined by having a BMI greater than or equal to 30.

**Figure 2 ijerph-20-07122-f002:**
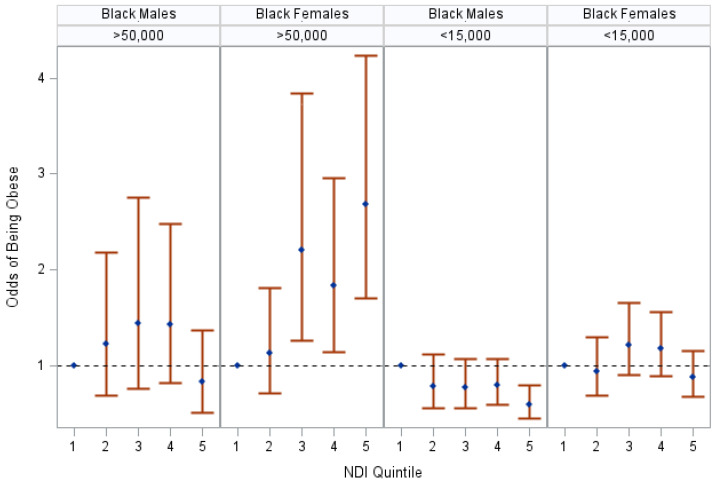
The association between NDI and obese status in Black participants, stratified by sex and household income, SCCS at baseline, 2002–2009. The figure displays odds ratios and 95% confidence intervals of being obese at baseline in comparison to “healthy” weight participants living in a neighborhood in the highest-SES quintile (Q1). Odds ratios were adjusted for age, insurance coverage, education, and smoking status. Stratifications were based on effect modification by household income in Black females (*p*-interaction = 0.001). Neighborhood SES is represented by the neighborhood deprivation index (NDI); “healthy” weight is defined by having a body mass index (BMI) of (18.5–24.9); obese status is defined by having a body mass index (BMI) greater than or equal to 30.

**Figure 3 ijerph-20-07122-f003:**
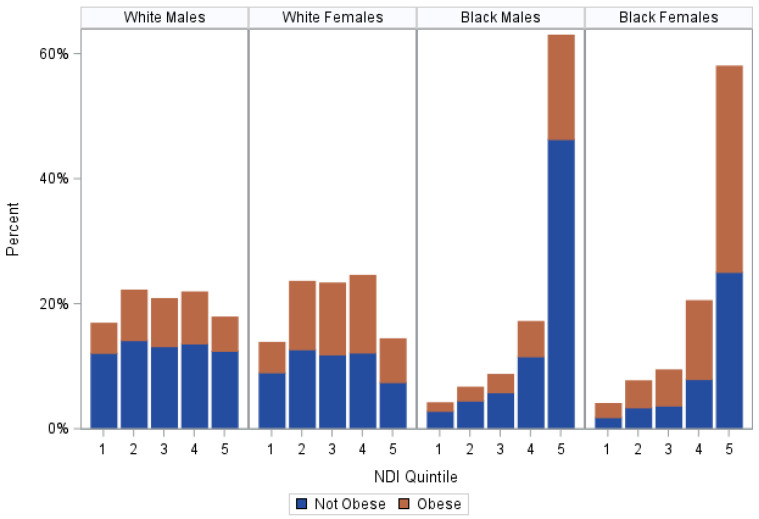
Percent of obese participants within each NDI quintile, stratified by race and sex, SCCS at baseline, 2002–2009. This figure shows the percentage of obese participants that live within each level of neighborhood SES at the time of their baseline interview in comparison to those of a different race and sex. In this study, White males most often resided in Q2 (22.25%, N = 2097), where 36.19% of them were obese, and White females most often resided in Q4 (24.61%, N = 3661), where 50.40% of them were obese. Black males most often resided in Q5 (63.04%, N = 13,975) along with Black females (58.09%, N = 18,041), where 26.46% and 56.77% of them were obese, respectively. NDI indicates neighborhood deprivation index; Q1 is the highest SES quintile and Q5 is the lowest SES quintile; and obese status is defined by having a BMI greater than or equal to 30.

**Table 1 ijerph-20-07122-t001:** Descriptive statistics from participants in the SCCS at baseline interview, 2002–2009; N = 80,970.

	Underweight, BMI < 18.5, N = 1017	“Healthy” Weight, 18.5 ≤ BMI ≤ 24.9, N = 19,704	Overweight, 25 ≤ BMI ≤ 29.9, N = 24,249	Obese, BMI ≥ 30, N = 36,000
Statistic (continuous)	Median (IQR ^b^)			
Age	50 (12.00)	50.00 (12.00)	51.00 (13.00)	51.00 (13.00)
BMI ^a^	17.64 (1.18)	22.76 (2.59)	27.44 (2.49)	35.15 (7.56)
Statistic (categorical)	N (row %)			
Sex				
Female	629 (1.31)	9062 (18.85)	12,442 (25.88)	25,936 (53.96)
Male	388 (1.18)	10,642 (32.35)	11,807 (35.89)	10,064 (30.59)
Race				
African American/Black	622 (1.17)	12,559 (23.60)	15,699 (29.49)	24,347 (45.74)
White	349 (1.44)	6327 (26.04)	7459 (30.70)	10,163 (41.83)
Other	46 (1.34)	818 (23.74)	1091 (31.67)	1490 (43.25)
Insurance				
Yes	593 (1.20)	10,956 (22.15)	14,954 (30.23)	22,961 (46.42)
No	424 (1.35)	8748 (27.77)	9295 (29.50)	13,039 (41.39)
Smoking status				
Current	732 (2.20)	11,729 (35.29)	10,317 (31.04)	10,460 (31.47)
Former	131 (0.71)	3191 (17.22)	5619 (30.32)	9592 (51.76)
Never	154 (0.53)	4784 (16.38)	8313 (28.47)	15,948 (54.62)
Enrolled at community health center	907 (1.30)	16,968 (24.30)	20,352 (29.15)	31,589 (45.25)
Household income				
<$15,000	735 (1.64)	11,766 (26.26)	12,618 (28.16)	19,686 (43.94)
$15,000–25,000	175 (1.01)	3774 (21.82)	5189 (30.01)	8155 (47.16)
$25,000–50,000	66 (0.59)	2285 (20.32)	3625 (32.23)	5271 (46.87)
>$50,000	41 (1.32)	1879 (52.83)	2817 (74.18)	2888 (71.67)
Education level				
Less than high school degree	388 (3.38)	5733 (48.02)	6611 (57.51)	10,423 (91.09)
High school degree/GED	329 (1.23)	6658 (24.88)	7915 (29.57)	11,862 (44.32)
Some college or technical/vocational degree	231 (1.12)	4720 (22.91)	6221 (30.20)	9426 (45.76)
College degree or more	69 (0.66)	2593 (24.81)	3501 (33.50)	4389 (41.04)
Deprivation quintile				
1 (least deprived)	75 (1.19)	1763 (28.03)	2134 (33.93)	2318 (36.85)
2	106 (1.04)	2341 (22.98)	3234 (31.75)	4504 (44.22)
3	130 (1.19)	2399 (21.95)	3281 (30.02)	5118 (46.83)
4	173 (1.03)	3558 (21.25)	4849 (28.96)	8163 (48.75)
5 (most deprived)	533 (1.45)	9643 (26.19)	10,751 (29.20)	15,897 (43.17)

^a^ BMI indicates body mass index; cutoffs based on designations for underweight, “healthy” weight, overweight, and obese. ^b^ IQR indicates interquartile range.

**Table 2 ijerph-20-07122-t002:** Associations between neighborhood deprivation and BMI, stratified by race and sex, among SCCS at baseline interview, 2002–2009.

	White Males			White Females			Black Males			Black Females		
Deprivation Index ^a^	BMI ^b^ Estimate ^c^	95% SE	*p*-Value	BMI ^b^ Estimate ^c^	95% SE	*p*-Value	BMI ^b^ Estimate ^c^	95% SE	*p*-Value	BMI ^b^ Estimate ^c^	95% SE	*p*-Value
Intercept ^d^	26.81	0.32	<0.001	28.48	0.34	<0.001	27.00	0.25	<0.001	30.11	0.30	<0.001
Q1	0.0 (ref)			0.0 (ref)			0.0 (ref)			0.0 (ref)		
Q2	0.93	0.20	<0.001	1.45	0.22	<0.001	0.06	0.24	0.80	0.32	0.27	0.24
Q3	0.82	0.21	<0.001	1.66	0.22	<0.001	0.23	0.23	0.31	0.69	0.26	0.01
Q4	0.81	0.21	<0.001	1.92	0.22	<0.001	0.35	0.21	0.10	0.81	0.24	0.00
Q5	0.09	0.22	0.68	1.48	0.25	<0.001	−0.21	0.20	0.28	0.47	0.23	0.04

^a^ Neighborhood deprivation quintiles, with Q1 being the least deprived and Q5 being the most deprived. ^b^ BMI= kg/m^2^ and indicates body mass index. ^c^ Analyses excluded participants with a BMI < 18.5 and were adjusted for age, health insurance status, education, household income, smoking status, and enrollment source. ^d^ Intercept represents the estimated BMI for participants aged 40–44 with an income <15 k, having no health insurance, with less than 9 years of education, who were current smokers, enrolled in a community health center, and living in the quintile for the highest-SES neighborhoods. The stratifications were based on the significant likelihood ratio test for interaction by race (*p*-interaction < 0.0001) and sex (*p*-interaction < 0.0001).

## Data Availability

The data presented in this study are available from the Southern Community Cohort Study upon request. The data are not publicly available to protect participant privacy.

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
