# Peer review of "Race- and Gender-Specific Associations between Neighborhood-Level Socioeconomic Status and Body Mass Index: Evidence from the Southern Community Cohort Study"

_ijerph, 2023, doi:10.3390/ijerph20237122_

Round 1

Reviewer 1 Report

Comments and Suggestions for Authors

My opinion is that this paper is presented in a very disorganized way which makes it hard to understand the important points. It´s not the type of data or the analysis, but the way the  results are presented, it is confusing.  The authors should present clearly what´s important, because otherwise it´s hard to understand it. 

Also, the title should be modified, because the evidence is clear on this association. Other results should be presented as title

Author Response

Authors’ Responses to Reviewers’ Comments

RE: ijerph-2676382, Giurini et al, Low Neighborhood-level Socioeconomic Status is Associated with Obesity: Evidence from the Southern Community Cohort Study

We thank the Editor and Reviewers for their helpful comments on our manuscript. We have considered the suggestions and revised our manuscript as requested. Our point-by-point responses to the specific comments are provided below. The page numbers referenced in the response section refer to the clean version of the manuscript with all changes accepted.

Reviewer 1 Comments and Suggestions for Authors

1: My opinion is that this paper is presented in a very disorganized way which makes it hard to understand the important points. It´s not the type of data or the analysis, but the way the results are presented, it is confusing.  The authors should present clearly what´s important because otherwise it´s hard to understand it. 

Response: Thank you for your comment. In order to clarify the findings, we have moved interpretation sentences from the Results to the Discussion section [now at line 245], edited table column headings [page 4, line 147] and added additional, relevant and more recent citations to support our results.

2: Also, the title should be modified, because the evidence is clear on this association. Other results should be presented as title

Response: We have modified the title to “Race and Gender-Specific Associations between Neighborhood Socioeconomic Status and Body Mass Index: Evidence from the Southern Community Cohort Study”.

Reviewer 2 Report

Comments and Suggestions for Authors

This study examined the association between neighborhood SES and body mass index. In addition, authors refer to obesity throughout the hall manuscript as a high BMI.

They only measure BMI as an index for obesity. However, it is well established that BMI a lone can’t be an accurate measurement for obesity. How authors can address this issue?

Authors used very old data 2002-2009. Why it took them 7 years to collect these data?

Why you are publishing it after about 15 years? Can the authors justify this issue?

Author Response

Authors’ Responses to Reviewers’ Comments

RE: ijerph-2676382, Giurini et al, Low Neighborhood-level Socioeconomic Status is Associated with Obesity: Evidence from the Southern Community Cohort Study

We thank the Editor and Reviewers for their helpful comments on our manuscript. We have considered the suggestions and revised our manuscript as requested. Our point-by-point responses to the specific comments are provided below. The page numbers referenced in the response section refer to the clean version of the manuscript with all changes accepted.

Reviewer 2 Comments and Suggestions for Authors

This study examined the association between neighborhood SES and body mass index. In addition, authors refer to obesity throughout the hall manuscript as a high BMI.

1: They only measure BMI as an index for obesity. However, it is well established that BMI alone can’t be an accurate measurement for obesity. How authors can address this issue?

Response: We agree with the Reviewer that BMI is an imperfect measure of obesity. We added information to the Discussion to describe the limitations of BMI as anthropometric variable [page 9 of 12, line 320]. Additionally, to address this limitation, we analyzed the BMI as a continuous variable and as a categorical variable. Modeling BMI continuously controls for error in categorical assignment.

2: Authors used very old data 2002-2009. Why it took them 7 years to collect these data?

Response: The Southern Community Cohort Study had a goal of recruiting over 85,000 participants into the cohort, and 7 years to enroll in the full sample partially attributable to the large sample size and the complex study design of enrolling participants at community health centers and through a stratified random sample of the general population.

3: Why you are publishing it after about 15 years? Can the authors justify this issue?

Response: We thank the Reviewer for the opportunity to clarify. Researchers from the Southern Community Cohort Study (SCCS) have not previously examined the association between neighborhood and obesity. The SCCS is a longitudinal study in which follow-ups have occurred approximately every 5 years since the baseline interviews in 2002-2009. In a future manuscript, we intend to examine the longitudinally between neighborhood and change in BMI, however, we would like to publish the current study showing baseline associations.  

Reviewer 3 Report

Comments and Suggestions for Authors

The study explores the relations between neighborhood-level SES and BMI with specific focus on overweight and obesity among individuals that are of low individual-level and neighborhood-level SES. Wide geographical coverage, large sample size, and a representative sample are the main strengths of the study. The fact that BMI, which is the independent variable of the study, was calculated based on self-report is the main limitation. The manuscript is well-written and provides a clear context. The reference list includes the relevant literature and in an unbiased manner. There is a need of revision in three points.

The classes must be non-overlapped. Any of the data does not fall into two different classes at once. This means that there must be a class for every data value in the dataset (e.g. less than $15,000; $15,000-25,000; $25,000-50,000; $50,000-100,000; $100,000 or more).

In Table 1, instead of "Healthy" weight 18.5<BMI<24.9", the second column heading should be "Healthy" weight 18.5≤BMI≤24.9".

In Table 1, instead of "Overweight 25<BMI<29.9", the third column heading should be "Overweight

25≤BMI≤29.9".

In Table 1, row percentages should be written instead of column percentages. Because there were approximately one and a half times as many women as men in the study group, presenting column percentages would mislead the reader.

In the findings section, only the data should be presented and no comments should be made. Therefore, the sentence that variation in the associations between neighborhood-level SES and BMI by demographic groups may be partially explained by differences in the prevalence of exposure and outcome by race and gender, should be moved to the discussion section.

One-third of the reference list consists of studies older than ten years. I recommend that the study be supported by new literature sources.

Author Response

Authors’ Responses to Reviewers’ Comments

RE: ijerph-2676382, Giurini et al, Low Neighborhood-level Socioeconomic Status is Associated with Obesity: Evidence from the Southern Community Cohort Study

We thank the Editor and Reviewers for their helpful comments on our manuscript. We have considered the suggestions and revised our manuscript as requested. Our point-by-point responses to the specific comments are provided below. The page numbers referenced in the response section refer to the clean version of the manuscript with all changes accepted.

Reviewer 3 Comments and Suggestions for Authors

The study explores the relations between neighborhood-level SES and BMI with specific focus on overweight and obesity among individuals that are of low individual-level and neighborhood-level SES. Wide geographical coverage, large sample size, and a representative sample are the main strengths of the study. The fact that BMI, which is the independent variable of the study, was calculated based on self-report is the main limitation. The manuscript is well-written and provides a clear context. The reference list includes the relevant literature and in an unbiased manner. There is a need of revision in three points.

1: The classes must be non-overlapped. Any of the data does not fall into two different classes at once. This means that there must be a class for every data value in the dataset (e.g. less than $15,000; $15,000-25,000; $25,000-50,000; $50,000-100,000; $100,000 or more).

Response: Thank you for the suggestion, we have revised the classes for the variables in Section 2.5 Statistical Analysis as follows: less than or equal to $14,999; $15,000-24,999; $25,000-49,999; $50,000-99,999; $100,000 or more [page 3 of 12, line 105].

2: In Table 1, instead of "Healthy" weight 18.5<BMI<24.9", the second column heading should be "Healthy" weight 18.5≤BMI≤24.9".

Response: We have revised the column heading in Table 1 for “Healthy” weight [page 4 of 12, line 147].

3: In Table 1, instead of "Overweight 25<BMI<29.9", the third column heading should be "Overweight 25≤BMI≤29.9".

Response: We have revised the column heading in Table 1 for Overweight [page 4 of 12, line 147].

4: In Table 1, row percentages should be written instead of column percentages. Because there were approximately one and a half times as many women as men in the study group, presenting column percentages would mislead the reader.

Response: We have changed the percentages in Table 1 to row percentages [page 4 of 12, line 147].

5: In the findings section, only the data should be presented and no comments should be made. Therefore, the sentence that variation in the associations between neighborhood-level SES and BMI by demographic groups may be partially explained by differences in the prevalence of exposure and outcome by race and gender, should be moved to the discussion section.

Response: Thank you for your thorough review, we agree and have removed this sentence from the Results section and moved it to the Discussion section [page 8 of 12, line 245].

6: One-third of the reference list consists of studies older than ten years. I recommend that the study be supported by new literature sources.

Response: We thank the Reviewer for the suggestion. We revised the citations to replace citations that occurred prior to 2010, and instead cite more recent references [see References, page 11 of 12, lines 403, 405, 431, and 433]. Several of the studies including Signorello 2005, 2010 and the Messer 2006 papers are critical to the methodology of this study, and therefore, we continue to include these citations.

Round 2

Reviewer 1 Report

Comments and Suggestions for Authors

The article is well organized now and more evidence has been included. The title describes correctly the purpose of the paper.  In line 230, you have to write obesity and delete BMI